# Identification of Brain-Specific Treatment Effects in NPC1 Disease by Focusing on Cellular and Molecular Changes of Sphingosine-1-Phosphate Metabolism

**DOI:** 10.3390/ijms21124502

**Published:** 2020-06-24

**Authors:** Anne Gläser, Franziska Hammerl, Markus H. Gräler, Sina M. Coldewey, Christin Völkner, Moritz J. Frech, Fan Yang, Jiankai Luo, Eric Tönnies, Oliver von Bohlen und Halbach, Nicola Brandt, Diana Heimes, Anna-Maria Neßlauer, Georg Christoph Korenke, Marta Owczarek-Lipska, John Neidhardt, Arndt Rolfs, Andreas Wree, Martin Witt, Anja Ursula Bräuer

**Affiliations:** 1Research Group Anatomy, School of Medicine and Health Sciences, Carl von Ossietzky University Oldenburg, 26129 Oldenburg, Germany; anne.glaeser@uni-oldenburg.de (A.G.); franziska.hammerl@uni-oldeburg.de (F.H.); nicola.brandt@uni-oldenburg.de (N.B.); 2Institute of Anatomy, Rostock University Medical Center, 18057 Rostock, Germany; dianaheimes@web.de (D.H.); anna-maria.nesslauer@uni-rostock.de (A.-M.N.); andreas.wree@med.uni-rostock.de (A.W.); martin.witt@med.uni-rostock.de (M.W.); 3Department of Anaesthesiology and Intensive Care Medicine, Center for Sepsis Control and Care (CSCC), Center for Molecular Biomedicine (CMB), Jena University Hospital, 07745 Jena, Germany; markus.graeler@med.uni-jena.de; 4Department of Anaesthesiology and Intensive Care Medicine, Septomics Research Center, Center for Sepsis Control and Care, Jena University Hospital, 07747 Jena, Germany; sina.coldewey@med.uni-jena.de; 5Translational Neurodegeneration Section “Albrecht Kossel”, Department of Neurology, University Medical Center Rostock, University of Rostock, 18147 Rostock, Germany; christin.voelkner@med.uni-rostock.de (C.V.); moritz.frech@med.uni-rostock.de (M.J.F.); timyang14@hotmail.com (F.Y.); Jiankai.luo@uni-rostock.de (J.L.); 6Center for Transdisciplinary Neurosciences Rostock (CTNR), Rostock University Medical Center, University of Rostock, 18147 Rostock, Germany; 7Institute of Anatomy and Cell Biology, University Medicine Greifswald, 17487 Greifswald, Germany; eric.toennies@med.uni-greifswald.de (E.T.); oliver.vonbohlen@uni-greifswald.de (O.v.B.u.H.); 8Department of Pediatrics, Klinikum Oldenburg, 26133 Oldenburg, Germany; korenke.christoph@klinikum-oldenburg.de; 9Human Genetics, School of Medicine and Health Sciences, University of Oldenburg, 26129 Oldenburg, Germany; marta.owczarek-lipska@uni-oldenburg.de (M.O.-L.); john.neidhardt@uni-oldenburg.de (J.N.); 10Junior Research Group, Genetics of childhood brain malformations, School of Medicine and Health Sciences, University of Oldenburg, 26129 Oldenburg, Germany; 11Research Center for Neurosensory Science, Carl von Ossietzky University Oldenburg,26129 Oldenburg, Germany; 12Centogene AG, D-18055 Rostock, Germany; arndt.rolfs@centogene.com

**Keywords:** Niemann–Pick disease type C1, brain, fibroblasts, S1P, sphingosine-1-phosphate receptors, qRT-PCR, sphingolipids, HPTLC, mass spectrometry, white matter

## Abstract

Niemann–Pick type C1 (NPC1) is a lysosomal storage disorder, inherited as an autosomal-recessive trait. Mutations in the *Npc1* gene result in malfunction of the NPC1 protein, leading to an accumulation of unesterified cholesterol and glycosphingolipids. Beside visceral symptoms like hepatosplenomegaly, severe neurological symptoms such as ataxia occur. Here, we analyzed the sphingosine-1-phosphate (S1P)/S1P receptor (S1PR) axis in different brain regions of *Npc1^−/−^* mice and evaluated specific effects of treatment with 2-hydroxypropyl-β-cyclodextrin (HPβCD) together with the iminosugar miglustat. Using high-performance thin-layer chromatography (HPTLC), mass spectrometry, quantitative real-time PCR (qRT-PCR) and western blot analyses, we studied lipid metabolism in an NPC1 mouse model and human skin fibroblasts. Lipid analyses showed disrupted S1P metabolism in *Npc1^−/−^* mice in all brain regions, together with distinct changes in *S1pr3*/S1PR3 and *S1pr5*/S1PR5 expression. Brains of *Npc1^−/−^* mice showed only weak treatment effects. However, side effects of the treatment were observed in *Npc1^+/+^* mice. The S1P/S1PR axis seems to be involved in NPC1 pathology, showing only weak treatment effects in mouse brain. *S1pr* expression appears to be affected in human fibroblasts, induced pluripotent stem cells (iPSCs)-derived neural progenitor and neuronal differentiated cells. Nevertheless, treatment-induced side effects make examination of further treatment strategies indispensable.

## 1. Introduction

Niemann–Pick disease type C1 (NPC1) is a rare autosomal-recessive lipid storage disease, which is caused by mutations in the *Npc1* gene (95% of the patients) [1,2]. It encodes for the transmembrane protein NPC1, found in late endosomes, that is suggested to be involved in the intracellular translocation of unesterified cholesterol to other cytoplasmic cell compartments [3,4]. Mutations result in impaired lipid trafficking, characterized by neurovisceral accumulation of unesterified cholesterol and glycosphingolipids, sphingosine (Sph), gangliosides (GM2, GM3) and other fatty acids in the endosomal/lysosomal system (LE/LY) [5,6,7]. This results in a heterogeneous, multisystemic spectrum of symptoms, such as extensive loss of Purkinje cells in the cerebellum (CE) and a variety of progressive neurological and visceral symptoms, such as ataxia, dystonia, dysphagia, psychiatric problems and hepatosplenomegaly as one of the first symptoms occurring [8,9,10,11,12,13,14].

The age of onset ranges from early infancy to an adolescent/adult onset corresponding to the estimated lifespan ranging from a few days to about 60 years [15,16]. To date, over 400 NPC1 mutations are known (www.hgmd.cf.ac.uk) [17,18,19]. The most common mutation, I1061 T, correlates with the classical juvenile phenotype, frequently found in patients with Western European descent or in Hispanic patients who originated from the Upper Rio Grande Valley in the U.S.A. The I1061 T substitution results in misfolding and subsequent degradation of the mutated NPC1 protein by the proteostasis machinery [20,21,22].

Currently, there is no cure for NPC1 patients. The only symptomatic therapy for NPC1, approved by the European Medicines Agency, is the iminosugar miglustat (*N*-butyldeoxynojirimycin, Zavesca^®^) [23]. The ability to cross the blood–brain barrier targets miglustat for treatment of neurological symptoms in NPC1 [24,25]. Miglustat inhibits glucosylceramide synthase, a key enzyme involved in the first step of glycosphingolipid synthesis, converting ceramides (Cer) to glucocerebrosides [26]. This substrate–reduction therapy reduces the intracellular accumulation of glycosphingolipids and results in a delayed progression of neurological symptoms and a prolonged life span [23,25,27,28]. Another promising drug is the cyclic-oligosacccharide 2-hydroxypropyl-β-cyclodextrin (HPβCD). It causes the release of cholesterol from LE/LY compartments (via a still unknown mechanism) and seems to be effective in NPC1 patients [29]. The treatment results in reduced intracellular cholesterol accumulation, and a later onset of neurological symptoms with an increased lifespan [30,31,32]. However, the therapy is associated with adverse side effects (e.g., hearing loss) [33]. Moreover, preliminary data of a multicenter, multinational phase 2b/3 clinical efficacy trial raised doubts concerning the benefit of the therapy [34]. With the aim of improving the therapeutic approach, a combination of miglustat, HPβCD and the neurosteroid allopregnanolone (pregnan-3alpha-ol-20-one) has been applied in animal models [12,30,35,36,37].

The *BALB/cNctr-Npc1^m1N^/J* mouse model used in this study shows a neurological phenotype with neurovisceral lipid accumulation of cholesterol and sphingolipids [38,39]. Former studies using this NPC1 mouse model and the combination treatment with miglustat, HPβCD and allopregnanolone showed alleviated lipid storage in numerous organs (e.g., liver, spleen, olfactory epithelium, CNS), improved olfactory performance via increased regeneration of the olfactory epithelium, reduced cerebellar Purkinje-cell loss and decreased motor dysfunction [12,30,35,36,37,40,41]. Normally, the efflux of sphingolipids like sphingosine from the LE/LY is supported by the NPC1 protein [42,43]. Sphingosine is phosphorylated by sphingosine kinases (SPHK) to generate sphingosine-1-phosphate (S1P), which acts extracellularly as a bioactive signaling molecule for five G-protein coupled receptors, called sphingosine-1-phosphate receptor 1–5 (S1PR1–5) [44]. S1P/S1PR1-5 interaction triggers intracellular signaling pathways mediated by Rho-/Ras, Phospholipase C (PLC), Phosphoinositide 3-kinase (PI3 K) and protein kinase B (Akt), modulating cell survival, proliferation, differentiation, inflammation and migration of neurons and glial cells in the central nervous system [45]. Impaired sphingosine trafficking results in changed S1P level in *Npc1^−/−^*mice, affecting additional pathways [40,42,43]. Speak et al. (2014) described an altered S1P level and distribution of natural killer cells in *Npc1^−/−^* mice identical to that in *S1pr5^-/-^* mice [43]. Furthermore, at the molecular level there is altered *S1pr* expression in spleen tissue of *Npc1^−/−^* mice that can be partially prevented by treatment with miglustat, allopregnanolone and HPβCD. However, the treatment causes side effects such as a reduced number of cytotoxic T lymphocytes (CTLs) and raised numbers of T helper (Th) cells [40].

The current study focused on the effects of treatment on various brain regions of *Npc1^−/−^* mice by investigating molecular and cellular changes in the S1P metabolism. Additionally, we found that *S1pr* expression was also changed in NPC1 patient-derived samples. Consequently, we used a combination of in vivo and in vitro approaches, including NPC1 patient-derived skin material and a transgenic mouse model.

## 2. Results

### 2.1. Disruption of Lipid Homeostasis in Different Brain Regions under Treatment

Previous studies have already shown an accumulation of phospho- and sphingolipid species in different tissues of *Npc1^−/−^* mice [46,47]. Especially the brain is strongly affected by the disturbed lipid metabolism. To identify these changes more clearly, we investigated the lipid profile of *Npc1^−/−^* mice in different regions of the brain and the effect of a protective treatment with miglustat, HPβCD and allopregnanolone via high-performance thin-layer chromatography (HPTLC, Figure 1). To better identify the lipid changes between the treatment groups, scanning profiles are also shown. Changed band intensities of the lipid classes are labeled with numbers in each brain region. TopFluor lysophosphatidic acid (LPA) was used as the internal standard, binding on the silica gel plate at retention factor (Rf) 0.13 and 0.57 (Appendix A).

#### 2.1.1. Parieto-Occipital Cortex (PC) 

The PC showed noticeable band differences in three lipid species (Figure 1A). Compared to sham-treated *Npc1^+/+^* mice, all of them showed weaker band intensities in sham-treated and treated *Npc1^-/-^* mice. However, treated *Npc1^−/−^* mice showed stronger band intensities compared to sham-treated *Npc1^-/-^* mice. There were no noticeable changes in band pattern between treated and sham-treated *Npc1^+/+^* mice (Figure 1A).

#### 2.1.2. Frontal Cortex (FC)

The FC showed equal changes of the band pattern in five lipid classes (Figure 1B). Compared to sham-treated *Npc1^+/+^* mice, there were weaker band intensities in sham-treated and treated *Npc1^−/−^* mice and stronger intensities in treated *Npc1^+/+^* mice. However, treated *Npc1^−/−^* mice showed stronger band intensities than sham-treated *Npc1^−/−^* mice (Figure 1B).

#### 2.1.3. Hippocampus (HC)

In the HC, five lipid species were changed (Figure 1C). Compared to sham-treated *Npc1^+/+^* mice, the following changes in band intensities were observed: sham-treated *Npc1^−/−^* mice showed weaker intensities of lipid classes 2 to 5 and stronger intensity in lipid class 1, treated *Npc1^−/−^* mice showed stronger intensities in lipid classes 2 to 5 and weaker intensities in lipid class 1. However, treated *Npc1^−/−^* mice showed slightly stronger intensities in lipid classes 2 to 4 and weaker intensities in lipid classes 1 and 5 compared to sham-treated *Npc1^−/−^* mice. There were no noticeable changes between sham-treated and treated *Npc1^+/+^* mice (Figure 1C).

#### 2.1.4. Cerebellum (CE)

The CE, as well as the BS, showed smaller changes in band intensities than the FC, PC and the HC (Figure 1D). Compared to sham-treated *Npc1^+/+^* mice all lipid classes showed weaker band intensities in sham-treated *Npc1^−/−^* mice and treated *Npc1^+/+^* mice. All lipid classes showed stronger band intensities in treated *Npc1^−/−^* mice compared to sham-treated *Npc1^−/−^* mice, whereby, lipid classes 2 to 5 showed even stronger band intensities than sham-treated *Npc1^+/+^* mice (Figure 1D).

#### 2.1.5. Brain Stem (BS)

Two lipid classes were slightly changed in the BS (Figure 1E). The most distinct difference was the stronger band intensity in sham-treated *Npc1^−/−^* mice compared to sham-treated *Npc1^+/+^* mice. The treatment of *Npc1^−/−^* mice resulted, in both lipid classes, in similar band intensities compared to sham-treated *Npc1^+/+^* mice (Figure 1E).

Summarizing, the HPTLC analysis identified changed band intensities of various lipid classes as effects of the different treatments and depending on the brain regions.

### 2.2. Characterization of Lipid Changes in Sham-Treated and Treated Npc1^−/−^ Mice

For identification of the changed lipid classes, mass spectrometry was performed (Figure 2). In this study, we focused on analyzing sphingolipids. All results are presented in ratios, compared to the sham-treated *Npc1^+/+^* mice that were set to 1. Red boxes indicate an increase of the lipid amount, green boxes a reduction. All changes in lipid amounts are presented as tendencies.

#### 2.2.1. Sphingomyelins (SM)

The SM amount was only slightly changed in the respective brain regions. Sham-treated *Npc1^−/−^* mice showed a slightly increased amount in most of the SM in the PC, FC, HC and the BS, compared to sham-treated *Npc1^+/+^* mice. The treatment of the *Npc1^−/−^* mice led only to slight improvements of these changes in the PC, FC and HC, but not in the BS. The CE revealed a clear increase in most of the SM in sham-treated *Npc1^−/−^* mice. This change was clearly improved by the treatment. The treatment of *Npc1^+/+^* mice led, especially in the PC, FC and HC, to slightly decreased SM amount.

#### 2.2.2. Sphingosine (Sph)

Sph showed a strong increase in the PC, FC, HC and CE of sham-treated *Npc1^−/−^* mice that seems to be prevented by the treatment. The BS showed almost no changes in sham-treated and treated *Npc1^−/−^* mice. Treated *Npc1^+/+^*mice showed only in the PC and BS a very small increase compared to sham-treated *Npc1^+/+^* mice.

#### 2.2.3. Sphingosine-1-Phosphate (S1P)

S1P level showed a very distinct decrease in sham-treated *Npc1^−/−^* mice in all brain regions. This effect was mostly not prevented by treatment; there was just a small increase of S1P in the PC, CE and BS of treated *Npc1^−/−^* mice compared to sham-treated *Npc1^−/−^* mice. The FC, however, showed a stronger protection effect by the treatment. Interestingly, the treatment of *Npc1^+/+^* mice showed a strong decrease of the S1P amount in all brain regions.

#### 2.2.4. Ceramides (Cer)

Cer showed clear changes in sham-treated *Npc1^-/-^* mice, with a reduction of Cer 18:1 in the PC, FC and HC, CE and BS showed almost no reduction in sham-treated *Npc1^−/−^* mice, there was, however, a strong increase of Cer 24:0, Cer 24:1 and Cer 26:1 in the CE. The treatment of *Npc1^+/+^* mice showed different effects. Most of the Cer in the PC, FC, HC and BS were reduced. PC and CE also showed a small increase of some Cer (Cer 24:1 in PC; Cer 24:1 and Cer 26:1 in CE). The treatment of *Npc1^−/−^* mice led to less pronounced reduction of Cer 18:1 in the PC, FC and HC. Likewise, the treatment led to increased Cer levels in the HC (Cer 26:1), CE (Cer 24:0, Cer 24:1, Cer 26:1) and BS (Cer 24:0, 26:1), compared to sham-treated *Npc1^+/+^*mice. Particularly the CE showed less pronounced changes in Cer 24:0, Cer 24:1 and Cer 26:1 by the treatment.

#### 2.2.5. Dihydro Ceramides (DC)

In sham-treated *Npc1^-/-^* mice, the DC lipid amount in the PC, FC, HC and BS tended to be reduced, especially in DC 18:1. The CE showed no changes in sham-treated *Npc1^−/−^* mice. The treatment of *Npc1^+/+^* mice mostly led to a small reduction of DC in all brain regions, but exclusively DC 22:0 showed a clear increase in the HC. The treatment in *Npc1^−/−^* mice resulted in small protective effects, e.g., DC 18:1 or DC 24:1 in the HC and DC 18:1 or DC 22:0 in the BS. Likewise, lipid changes tended to be prevented in DC 18:1 and DC 24:1 in the PC and DC 18:1 in the FC by the treatment.

#### 2.2.6. Monohexosyl Ceramides (MC)

The lipid amount in most of the MC was reduced in the PC, FC and HC of sham-treated *Npc1^−/−^* mice. Sham-treated *Npc1^−/−^* mice showed no clear changes in the CE. In the BS only MC 26:0 was reduced in sham-treated *Npc1^−/−^* mice. The treatment of *Npc1^+/+^* mice led to a clear increase of MC 16:0 and MC 18:0 in the PC and FC. The HC showed an increase of MC 18:0 and a reduction of MC 18:1 only in treated *Npc1^−/−^* mice. The CE showed a reduced lipid amount in *Npc1^+/+^* mice of MC 18:1 and MC 26:0. The treatment of *Npc1^−/−^* mice led to a clear increase of the lipid amount, especially of MC 16:0 and MC 18:0 in the PC and FC. In the HC, the MC 18:0 showed similar changes as in the PC and FC. However, the treatment of *Npc1^−/−^* mice tended to be effective in most of the MCs in the PC and the HC (except MC 26:0 in the HC).

#### 2.2.7. Monohexosyl Dihydroceramides (MDC)

MDC showed the same changes as the respective MC.

#### 2.2.8. Lactosylceramides (LC)

Most of the LC were increased, especially in PC and FC of sham-treated *Npc1^−/−^* mice, whereas the HC showed just a small increase in the LC 16:0 and LC 18:1 amount, but a distinct decrease in LC 24:1. The treatment of *Npc1^+/+^* mice caused less smaller changes in the PC, FC, CE and BS, but distinct decreases of most LC in the HC. The increased lipid amount of sham-treated *Npc1^−/−^* mice in the PC, FC and HC tended to be prevented by treatment. CE and BS showed no clear changes of LC amounts in any brain regions.

Summarizing, the treatment prevented changes in some lipid classes. Lipid classes of some brain regions, however, showed no protective effect and in some regions side effects occurred.

For further studies, we focused on S1P, a ligand of five different S1P receptors (S1PR1–5), as the only lipid showing no protective treatment effects in *Npc1^−/−^* mice, as well as showing side effects in *Npc1^+/+^* mice in all brain regions.

### 2.3. Distinct Changes of S1pr3 and S1pr5 mRNA Expression in Npc1^−/−^ Mice

To investigate whether the disturbed lipid metabolism is associated with molecular events, various lipid receptors were analyzed. Due to the demonstrated changes of the sphingolipids in different brain regions of sham-treated and treated *Npc1^+/+^* and *Npc1^−/−^* mice, we focused on analyzing the *S1prs* in the same brain regions of all treatment groups using quantitative real-time PCR (qRT-PCR, Figure 3). *S1pr1*, *S1pr3,* and *S1pr5* are the most strongly expressed receptors in all brain regions, whereby the *S1pr3* and the *S1pr5* receptors showed the strongest changes between the different groups (sham-treated *Npc1^+/+^* and *Npc1^−/−^*, treated *Npc1^+/+^* and *Npc1^−/−^*), depending on the brain region.

#### 2.3.1. Parieto-Occipital Cortex (PC)

The PC exhibited no differences in *S1pr1*, *S1pr3* and *S1pr4* expression in all groups (Figure 3A). The *S1pr2* expression showed no changes in sham-treated *Npc1^−/−^* mice. Nevertheless, the treatment of *Npc1^−/−^* resulted in a slight, but significant, increase compared to treated *Npc1^+/+^* mice. The *S1pr5* expression was significantly decreased in sham-treated *Npc1^−/−^* mice compared to the sham-treated and treated *Npc1^+/+^* mice. Likewise, treated *Npc1^−/−^* mice exhibited a significantly decreased *S1pr5* expression compared to sham-treated and treated *Npc1^+/+^* mice (Figure 3A).

#### 2.3.2. Frontal Cortex (FC)

The FC showed small changes in *S1pr1* expression, with a discreet, but significant, increase in treated *Npc1^+/+^* mice compared to sham-treated *Npc1^-/-^* mice (Figure 3B). There were no other significant changes in *S1pr1* expression. *S1pr2* expression was significantly increased in sham-treated *Npc1^-/-^* mice compared to sham-treated and treated *Npc1^+/+^* mice. Equally, there was still a significant increase of treated *Npc1^-/-^* mice compared to sham-treated and treated *Npc1^+/+^* mice. *S1pr3* showed a slight, but significant, increase in treated *Npc1^-/-^* mice compared to the untreated control. There were no other significant changes in *S1pr3* expression. *S1pr4* expression showed no changes in sham-treated and treated *Npc1^+/+^* and *Npc1^-/-^* mice. *S1pr5*, however, exhibited similar changes in all groups as in the PC. Sham-treated *Npc1^-/-^* showed significantly decreased *S1pr5* expression compared to sham-treated and treated *Npc1^+/+^* mice. Treated *Npc1^-/-^* mice, however, still showed similar level to sham-treated *Npc1^-/-^* mice and a significant decrease compared to sham-treated and treated *Npc1^+/+^* mice (Figure 3B).

#### 2.3.3. Hippocampus (HC)

The HC presented no significant changes in *S1pr1* and *S1pr2* expression (Figure 3C). Nevertheless, *S1pr3* did show a slightly increased expression in treated *Npc1^-/-^* mice compared to sham-treated *Npc1^+/+^* mice. *S1pr4* expression was slightly decreased in sham-treated *Npc1^-/-^* mice compared to the sham-treated control. Treatment of *Npc1^+/+^* mice led to a significant decrease compared to the sham-treated control. *S1pr5* mRNA expression showed similar tendencies in all groups as in FC and PC. There, sham-treated *Npc1^-/-^* mice expression was decreased compared to sham-treated and treated *Npc1^+/+^* mice. Treated *Npc1^-/-^* mice showed a clear tendency to decreased expression compared to treated *Npc1^+/+^* mice and a still significant decrease compared to sham-treated *Npc1^+/+^* mice (Figure 3C).

#### 2.3.4. Cerebellum (CE)

In the CE there were no significant changes in *S1pr1*, *S1pr4* and *S1pr5* expression (Figure 3D). Nevertheless, *S1pr2* showed a slight effect in treated *Npc1^-/-^* mice, with a significantly increased expression compared to the treated control, as well as to the sham-treated *Npc1^-/-^* and *Npc1^+/+^* mice. *S1pr3* expression was significantly increased in sham-treated *Npc1^-/-^* mice compared to sham-treated and treated *Npc1^+/+^* mice. Likewise, treated *Npc1^-/-^* mice showed a clearly significant increase compared to sham-treated and treated *Npc1^+/+^.* The treatment of *Npc1^-/-^* mice also led to a significant decrease of *S1pr3* expression compared to sham-treated *Npc1^-/-^* mice (Figure 3D).

#### 2.3.5. Brain stem (BS)

The BS exhibited no changes in *S1pr1*, *S1pr2* and *S1pr4* expression in all groups (Figure 3E). *S1pr3* expression showed a significant increase in sham-treated *Npc1^-/-^* mice compared to sham-treated and treated *Npc1^+/+^* mice. Interestingly, as in the CE, the treatment resulted in a significantly decreased expression in *Npc1^-/-^* mice compared to sham-treated *Npc1^-/-^* mice*. S1pr5* showed significantly increased expression in treated *Npc1^+/+^* mice compared to sham-treated *Npc1^+/+^* and *Npc1^-/-^* mice (Figure 3E).

Expression data (*n* = 3) are shown as mean ± standard error of the mean (SEM). Corresponding p values are shown in Appendix A.

Due to the most prominent changes in *S1pr3* and *S1pr5* mRNA expression, further studies focused on S1PR3 protein expression in the BS and CE and on S1PR5 protein expression in the PC, FC and HC.

### 2.4. Changes of S1PR3/5 Protein Expression in Treated Npc1^-/-^ Mice

To check the effects of changed *S1pr3/5* mRNA expression on the protein level, western blot analyses were performed. To confirm the specificity of S1PR5 and S1PR3 antibody, GFP-coupled *mS1pr1-5* constructs were first cloned. Digestion with specific restriction enzymes confirmed the cloned constructs shown in Appendix A (pEGFP-*mS1pr1*, pEGFP-*mS1pr2*, pEGFP-*mS1pr3*, pEGFP-*mS1pr4*, pEGFP-*mS1pr5*). Subsequently, the functionality of the GFP-coupled *mS1pr1-5* constructs was shown by immunocytochemistry (Appendix A). Purified GFP-coupled S1PR1-5 protein lysates were used to show the specificity of the S1PR3 and S1PR5 antibody. The S1PR3 and S1PR5 antibody was exclusively detected in the respective GFP-coupled S1PR protein lysates (Appendix A). Quality and quantity of the lysates were checked using a GFP antibody, detecting respective posttranslational modified forms of all GFP-coupled S1PR1-5 proteins. Quantification of S1PR3 and S1PR5 in different brain regions of sham-treated and treated *Npc1^+/+^* and *Npc1^-/-^* mice followed.

S1PR5 protein expression analyses focused on the PC, FC and the HC, due to the changed *S1pr5* mRNA expression in these same regions, as previously described (Figure 4A–C). The PC and the FC showed the same tendencies as seen in *S1pr5* mRNA expression. Both regions tended to slightly decreased S1PR5 protein expression in the *Npc1^-/-^* sham, that tended to be prevented by the treatment. Nevertheless, both regions tended to an increase of S1PR5 protein expression in treated *Npc1^+/+^* mice, suggesting potential treatment effects. Exclusively, treated *Npc1^+/+^* mice showed a significant increase in the FC, compared to sham-treated *Npc1^-/-^* mice. Compared to the respective mRNA data, the HC showed different tendencies in S1PR5 protein expression. There were no changes of S1PR5 protein expression between sham-treated *Npc1^-/-^* and *Npc1^+/+^* mice. The treatment of *Npc1^+/+^* mice had no effect. *Npc1^-/-^* mice tended to show a slightly increased protein expression upon treatment.

Due to the previously described indications in *S1pr3* mRNA expression, the analyses of S1PR3 protein expression focused on the CE and BS (Figure 4D–E). The CE showed a significant increase in sham-treated *Npc1^-/-^* mice compared to sham-treated *Npc1^+/+^* mice, which could be prevented in treated *Npc1^-/-^* mice. Treated *Npc1^-/-^* mice showed significantly decreased S1PR3 protein expression compared to sham-treated *Npc1^-/-^* mice. *Npc1^+/+^* mice showed no distinct changes by the treatment; nevertheless, it was significantly decreased compared to sham-treated *Npc1^-/-^* mice. The BS showed similar tendencies as the CE. Sham-treated *Npc1^-/-^* mice expression levels tended to be increased compared to sham-treated *Npc1^+/+^* mice. This increase tended to be prevented by treatment. Treated *Npc1^+/+^* mice showed no distinct changes, but there was a significant reduction compared to sham-treated *Npc1^-/-^* mice.

In conclusion, the treatment had a prominent effect on S1PR3 protein expression, whereas S1PR5 protein expression tended to only show alterations in the brain regions PC and FC.

### 2.5. Cell-Type-Specific mRNA Expression of S1PRs

To analyze how the changed *S1pr* expression in *Npc1^-/-^* mice affected cellular events, initially cell-type-specific localization of *S1pr1-5* mRNA in mouse primary cells were investigated via qRT-PCR analysis (Appendix A). *S1pr1* and *S1pr2* showed high expression in astrocytes, neurons, microglia and oligodendrocytes compared to all other *S1prs*. *S1pr3* was the highest expressed receptor in astrocytes; it was, however, also expressed in neurons and oligodendrocytes. *S1pr3* had a very low expression in microglia. *S1pr4* was expressed in microglia, neurons and oligodendrocytes, but showed a very low expression in astrocytes. The *S1pr5* expression in astrocytes, microglia and neurons was very low or below the detection level. Interestingly, a strong *S1pr5* expression in oligodendrocytes was shown.

### 2.6. Treatment Did Not Prevent Reduction of the Diameter of the Corpus Callosum

This fact—and the very distinct change in *S1pr5* expression in sham-treated and treated *Npc1^-/-^* mice compared to sham-treated *Npc1^+/+^*mice—suggested performing morphometric analyses in white matter areas, such as corpus callosum (Figure 5A). Measuring the diameter confirmed the - already shown - reduction of the corpus callosum in *Npc1^-/-^* mice [48,49], but the treatment of *Npc1^-/-^* mice did not prevent the reduction of the corpus callosum. It was still significantly reduced compared to sham-treated *Npc1^+/+^* mice. Additionally, transmission electron microscopy of sham-treated *Npc1^-/-^* mice showed impaired nerve fibers, that were loaded with autophagosomes and defects in myelination in the corpus callosum, as previously shown by Bräuer et al. [34]. To examine whether the reduced corpus callosum and myelination defects are correlated, immunohistochemical analyses with 2´,3´-cyclic-nucleotide 3´-phosphodiesterase (CNPase), a marker for myelinated fibers and oligodendrocytes, were performed (Figure 5B–C). Interestingly, this study showed the same tendencies in sham-treated and treated *Npc1^-/-^*and *Npc1^+/+^* mice (no statistical analyses were performed due to small number of mice), strengthening the assumption that changes in *S1pr5* mRNA expression in oligodendrocytes and the decrease of corpus callosum diameter are correlated.

### 2.7. Is the S1pr1–5 Expression Affected in Human NPC1 Mutant Fibroblasts?

For the first time, *S1pr1–5* expression analyses in human NPC1-deficient fibroblasts were performed to evaluate whether the alterations in *Npc1^-/-^* mice are also detectable in human patient-derived cell lines. Patient-derived NPC1-deficient fibroblasts of two affected siblings (sibling 1, sibling 2, Figure 6A), both carrying the same NPC1 mutation (c.3182 T>C), were used to analyze the *S1pr1-5* expression at different passaging numbers in cell culture procedures (Figure 6B). Both patients were treated with miglustat for about twelve years before skin biopsy and culturing the fibroblasts. In the expectation that miglustat would be washed out through repeating splitting procedures, fibroblasts were cultivated for 15 splitting procedures (p), analyzing the *S1pr* expression at p1, p5, p10 and p15. First, filipin staining (Appendix A) showed an accumulation of cholesterol in NPC1-deficient fibroblasts (p5 of sibling 1 and 2) and fibroblasts of an apparently healthy individual, obtained from Coriell Institute for Medical Research (control: GM08398). Fibroblasts of sibling 1 and sibling 2 showed a distinct clustered accumulation of cholesterol compared to the homogeneously distributed cholesterol of the control. To examine the expression of *S1pr1-5* mRNA over 15 splitting procedures, qRT-PCR analyses revealed different tendencies of each *S1pr* mRNA in both patients. Expression data were not compared to fibroblasts of an apparently healthy family member because these were not available. Fibroblasts of another apparently healthy individual were not used, because no information about treatment and other diseases was available. Sibling 1 showed slight tendencies of increased *S1pr1* mRNA at p5, compared to p1, but during the processes it tended to be decreased. *S1pr2* and *S1pr4* also tended to decreased mRNA expression during the process of additive splitting procedures. However, *S1pr3* and *S1pr5* mRNA tended to be increased during the splitting processes. Sibling 2, who is less affected (weaker symptoms) than sibling 1, showed similar effects, but only for some receptors (*S1pr1*, *S1pr2*). *S1pr1* mRNA tended to an increase in p5, compared to p1, but tended to be decreased during the following splitting procedures. *S1pr2* mRNA tended to be reduced during the splitting processes. *S1pr3* tended to be decreased at p5, compared to p1 and tended to be increased during subsequent splitting procedures. *S1pr4* and *S1pr5* showed different tendencies in sibling 2 compared to sibling 1. Initially, *S1pr4* mRNA tended to be reduced until p10, but then to be increased at p15 (compared to p10). *S1pr5* mRNA tended to be strongly decreased at p5, p10 and p15 (compared to p1). Interestingly, the *S1pr5* mRNA expression level of sibling 2, the less affected patient, is clearly higher than in sibling 1. In summary, all *S1prs* are expressed in human fibroblast and showed variations, depending on the severity of the disease.

### 2.8. Altered S1pr5 Expression in Human Cells Derived from Induced Pluripotent Stem Cells (iPSCs)

In another experiment, we used patient-specific fibroblasts (control: GM05659; *Npc1* homozygous mutated: GM18453, c.3182 T>C), iPSC-derived neural progenitor cells (NPCs) and neuronal differentiated cells (NDCs) (Appendix A) to analyze *S1pr1-5* mRNA expression. Unfortunately, there is no additional information about the patient carrying the NPC1 mutation. Interestingly, the fibroblasts showed a significant reduction of *S1pr1* and *S1pr2* mRNA and a significant increase in *S1pr5* mRNA. *S1pr3* mRNA was not changed and *S1pr4* mRNA tended to be reduced. iPSC-derived NPCs showed the opposite of the fibroblasts, with a significant reduction of *S1pr5* mRNA. *S1pr1*, *S1pr2*, *S1pr3* and *S1pr4* showed now significant changes. iPSC-derived NDCs showed similar changes to NPCs with a significant reduction of *S1pr5* mRNA. Additionally, NDCs showed an increase of *S1pr1* mRNA. *S1pr2*, *S1pr3*, *S1pr4* now showed significant changes. Summarizing, *S1pr5* mRNA showed the most pronounced changes in the different cell types. The fibroblasts showed a strong increase, whereas the NPCs and NDCs showed contrary changes, with a strong reduction in *S1pr5* mRNA. *S1pr5* presents the most interesting changes in human NPC1 mutant fibroblasts and iPSC-derived cells.

## 3. Discussion

In the current study, using quantitative molecular and cellular techniques in the brains of *Npc1^-/-^* mice, we showed for the first time that S1P metabolism changes after treatment with miglustat, allopregnanolone and HPβCD. The structural and functional heterogeneity of the brain motivated us to investigate treatment effects in different brain regions (parieto–occipital cortex, frontal cortex, hippocampus, cerebellum and brain stem). Based on previous studies of treatment effects in the spleen [40], liver [36] and the olfactory system [35,41], our data partly confirmed the protective effects in the brain, but not to the same extent. Interestingly, our analysis at the molecular level demonstrated side effects in treated *Npc1^+/+^* mice that were also shown in previous studies at the cellular level of the spleen, showing a changed Th cell/CTLs ratio [40].

### 3.1. Weak Improvement of the S1P Metabolism in Treated Npc1^-/-^ Mice

Previous studies from Neßlauer et al. [40], Ebner et al. [36] and Meyer et al. [35,41] investigated the effects of the treatment in spleen, liver and the olfactory system, resulting mostly in cellular and molecular improvements of the NPC1 characteristics. However, our investigations of the treatment effects in the brain did not confirm protection effects of the same magnitude. Depending on the brain region, some lipid classes were differently protected by treatment. However, S1P was the only lipid that presented reduced lipid levels in all brain regions of sham-treated and treated *Npc1^-/-^* mice. Expression analysis of the associated *S1prs* revealed similar effects in most brain regions. *S1pr3*/S1PR3 and *S1pr5*/S1PR5 underwent the most prominent changes in *S1pr*/S1PR expression. *S1pr3*/S1PR3 was significantly increased in CE of sham-treated *Npc1^-/-^* mice. It is involved in neuroinflammatory gene expression in astrocytes, mediated by RhoA signaling [50]. Previous studies showed cerebellar astrogliosis in *Npc1^-/-^* mice [51,52]. These findings led to the assumption that an *S1pr3*/S1PR3 increase may play a role in astrocyte activation in *Npc1^-/-^* mice. Interestingly, the treatment showed a significant effect at the mRNA expression level—and at the protein level—it also seemed to show a protective effect, suggesting that the treatment may act on different regulation levels. The opposite effect was revealed by *S1pr5* expression. It was significantly decreased in the PC, FC and HC, responding under treatment only with small changes. *S1pr5*/S1PR5 is mainly expressed in oligodendrocytes, occurring extensively in the white matter. It is associated with oligodendrocyte proliferation, survival and inhibition of migration of oligodendrocyte precursor cells [53,54]. More detailed studies need to be conducted regarding the treatment effects on *S1pr*/S1PR expression and the entire S1P homeostasis of different brain regions.

### 3.2. Weak Treatment Effects at the Cellular Level in the White Matter

Metabolites of the sphingolipid metabolism, like MC and SM, are major components in the white matter, that are interesting especially regarding neurodegenerative diseases like NPC1 or multiple sclerosis [46,55,56,57,58]. Our lipid analysis in *Npc1^-/-^* mice confirmed changes in MC, SM and SM metabolites, whereby SM and Sph revealed increased lipid levels and S1P decreased levels in the different brain tissues. Previous studies of human NPC1-deficient fibroblasts [59] and human plasma samples [46] of NPC1 patients also showed reduced S1P levels. However, some previous studies also reported increased S1P levels in spleen [40], lymph nodes [43] and total brain of *Npc1^-/-^* mice [42,46], that are not consistent with our results. Possible reasons for this need to be investigated.

S1P acts as a signaling molecule for five S1PRs that are abundant in the brain [60]. Our expression analyses in all brain regions of *Npc1^-/-^* mice showed a distinct change of *S1pr3* and *S1pr5* expression, depending on the brain regions (an *S1pr3* increase in CE and BS and a *S1pr5* decrease in PC, FC and HC). Further analysis identified expression of *S1pr3* in astrocytes, oligodendrocytes and neurons, whereby *S1pr5* was mainly expressed in oligodendrocytes. Increased *S1pr3* mRNA expression was also demonstrated in brain tissue in experimental autoimmune encephalomyelitis (EAE), a model of multiple sclerosis [61,62] and in a mouse model of Sandhoff disease, another lipid storage disease that showed a milder course with reduced proliferation of glial cells and less astrogliosis in *S1pr3^-/-^* mice suffering from Sandhoff disease. This suggested that *S1pr3* may contribute to neurodegenerative diseases [63,64]. S*1pr5* plays a role in oligodendrocyte differentiation. Jaillard et al. [53] showed that *S1pr5* mediates two functional pathways, depending on the developmental stage of the oligodendroglial cells. On one hand, it mediates process retraction in immature oligodendrocytes and on the other hand it promotes survival in mature oligodendrocytes, but not in pre-oligodendrocytes.

Based on the myelination defects in *Npc1^-/-^* mice [34,48,49], we suggested that changes in oligodendrocyte-expressed *S1prs* may be correlated with white matter/myelinization defects. The measurement of the corpus callosum diameter confirmed previous data of reduced corpus callosum diameter in *Npc1^-/-^* mice. Interestingly, we found that the treatment had no effects in *Npc1^-/-^* mice. Furthermore, immunohistochemical analysis using a marker for myelinated fibers revealed the same tendencies in sham-treated and treated *Npc1^-/-^* mice when comparing morphometric measurements of the corpus callosum. These results suggested a correlation between changed *S1pr* expression and myelinization/white matter defects and showed that the treatment has no protective effects on white matter.

### 3.3. Side Effects of the Treatment in Npc1^+/+^ Mice

To date, NPC1 disease is not curable. Miglustat, the only approved drug in Europe, is only used for symptomatic treatment, especially reducing glycosphingolipid accumulation in the LE/LY system [28]. Likewise, HPβCD, mediating the excretion of cholesterol in *Npc1^-/-^* mice, represents another promising drug [29]. Clinical trials by Matsuo et al. [31] and Ory et al. [65] in NPC1 patients seemed to be effective, suggesting HPβCD as a potential therapy for NPC1 patients. However, a current multinational phase 2b/3 clinical study has raised doubts concerning the positive effects of HPβCD [34]. In addition, HPβCD, as well as miglustat, have side effects, such as tremor, weight loss or gastrointestinal problems induced by miglustat [66] and hair loss induced by HPβCD [33]. As shown in previous studies, there was side effects of the treatment at the cellular level in the spleen [40], showing a decreased number of CTLs and an increased number of Th cells. The treatment used in our study was also associated with side effects on S1P metabolism. Our present study, analyzing treatment effects in the brain, presented side effects on the S1P level in all brain regions with a strongly decreased S1P lipid amount. Beside S1P, other lipids showed changes, such as a strong increase of MC 18:0 and MDC 18:0 in the PC and FC and Cer 24:1 and Cer 26:1 in the CE or a decrease of LC 16:0 and LC 22:0 in the HC. Whether these effects are caused by miglustat or HPβCD needs to be investigated. This demonstrated the need to examine further promising treatments. The group of Sarah Spiegel [67] demonstrated that S1P functions as an endogenous inhibitor of class1/2 histone deacetylases (HDAC). Fingolimod (FTY720), a drug approved for treatment of multiple sclerosis, is a sphingosine analog and able to enter the nucleus. The active phosphorylated form (FTY720-P) accumulates in the CNS and also functions as an HDAC inhibitor to regulate gene expression [68]. Current studies showed increased NPC1 and NPC2 expression in fibroblasts of patients with NPC1 after FTY720 treatment, combined with decreased cholesterol and sphingolipid accumulation [69]. The accumulation in the CNS and the sole regulation of genes restricted to sphingolipid metabolism identified FTY720 as a promising treatment for NPC1 patients with neurological symptoms.

## 4. Materials and Methods

### 4.1. Animals

Heterozygous breeding pairs of NPC1 mice (BALB/cNctr-*Npc1^m1 N^*/J) were obtained from Jackson Laboratories (Bar Harbor, ME, USA) for generating homozygous *Npc1^-/-^* mutant mice and *Npc1^+/+^* control wild-type mice. Transgenic mice at postnatal day 65 (P65) were used for RNA, protein and lipid analyses, as well as mass spectrometric analysis and measurement of corpus callosum diameter. For preparation of oligodendrocytes, postnatal BALB/c mice were obtained from the central animal facility of the University Medical Center Rostock. Approval of the experiments was obtained from the “Landesamt für Landwirtschaft, Lebensmittelsicherheit und Fischerei Mecklenburg-Vorpommern” (May 2016: LALLF: 7221.3-1.1-011/16, June 2012: 7221.3-1.1-030/12).

For preparation of astrocytes, microglia and neurons, timed-pregnant and postnatal wild type C57BL/6 mice were obtained from the central animal facility (FEM) of the Charité—Universitätsmedizin Berlin. Approval of the experiments was obtained from the „Landesamt für Gesundheit und Soziales Berlin“(April 2011: LAGeSO: T0108/11).

Mice were maintained under standard laboratory conditions with 12 h day/night cycle, a temperature of 22 °C, a relative humidity of 60% and free access to food and water. This study was carried out in accordance with German and European guidelines (2010/63/EU) for the use of laboratory animals.

### 4.2. Genotyping

Genotyping of NPC1 mice until P7 was performed by PCR analysis. Direct PCR tail with proteinase K were used for homogenization of 1–2 mm cut tail, incubating at 55 °C for 16 h overnight with agitation (750 rpm, ThermoMixer^®^ C, Eppendorf, Hamburg, Germany). Lysates were centrifuged at 2400× *g* for 30 s. PCR analysis was performed using 2 µL lysate and two different primer pairs under equal cycling conditions. To detect the 475 bp fragment of the mutant allele, the following primer pair was used: 5′-ggtgctggacagccaagta-3′ and 5′-tgagcccaagcataactt-3′. To detect the 173 bp fragment of the wildtype allele the primers were: 5′-tctcacagccacaagcttcc-3′ and 5′-ctgtagctcatctgccatcg-3′.

### 4.3. Pharmacologic Treatment

*Npc1^+/+^* and *Npc1^-/-^* mice were treated with a combination of allopregnanolone (Sigma-Aldrich, St. Louis, MO, USA), HPβCD (Sigma-Aldrich) and miglustat (Actelion Pharmaceuticals, Allschwil, Switzerland), further referred to as “treated”. Control *Npc1^+/+^* and *Npc1^-/-^* mice received Ringer’s solution (B. Braun Melsungen AG, Melsungen, Germany) or 0.9% normal saline solution (Carl Roth GmbH, Karlsruhe, Germany) and are further referred to as “sham-treated”. The treatment scheme (Appendix A) started at P7 with a weekly intraperitoneal injection of allopregnanolone (25 mg/kg) dissolved in HPβCD (4000 mg/kg, dissolved in Ringer’s solution) as previously described [30,35,36,40]. Additionally, mice received a daily intraperitoneal injection of miglustat (300 mg/kg, dissolved in 0.9% normal saline solution) starting at P10 until P22. Afterwards, daily uptake was ensured by mixing miglustat powder (1200 mg/kg) with standard chow until P65. Sham-treated mice followed the same treatment scheme, receiving Ringer’s solution (B. Braun Melsungen AG) or 0.9% normal saline solution (Carl Roth GmbH).

### 4.4. Tissue Sampling

Sham-treated and treated *Npc1^+/+^* and *Npc1^-/-^* mice were deeply anesthetized with pentobarbital (90 mg/kg body weight, AbbVie, Berlin, Germany) and decapitated at P65. Subsequently, different brain regions were dissected: cortex, hippocampus (HC), cerebellum (CE) and brain stem (BS). The cortex was divided into frontal and parieto–occipital parts, referred to as frontal cortex (FC) and parieto–occipital cortex (PC). All brain regions were snap-frozen in liquid nitrogen and stored at −80 °C. The non-perfused tissue was used for HPTLC, mass spectrometry, qRT-PCR and western blot analyses.

### 4.5. Lipid Extraction

Lipids of different brain regions of sham-treated and treated *Npc1^+/+^* and *Npc1^-/-^* mice, stored at −80 °C, were extracted according to Bligh and Dyer (1959) [70], with slight modifications. The tissues were weighed and subsequently homogenized with the PT 3100D homogenizer (Kinematica AG, Luzern, Switzerland) in a mixture of chloroform, methanol and 37% hydrochloric acid (ratio 2:4: 0,1; all from Merck KGaA, Darmstadt, Germany) supplemented with 140 µL of 1% butylated hydroxytoluene (Merck KGaA), to prevent lipid oxidation. Then, 1.25 mL chloroform, 1.25 mL HPTLC water (Carl Roth GmbH) and 5 µL/50 mg brain tissue of the fluorescent standard TopFluor LPA (810280P, Sigma-Aldrich) were added to the tissue and incubated at room temperature (RT) for 30 min. TopFluor LPA was used as internal standard, ensuring the analysis of the same lipid amount in all compared samples (Appendix A) [71]. Samples were centrifuged at 1260× *g* for 10 min, resulting in a triphasic separation. The bottom phase, containing a mixture of chloroform and lipids, was transferred into a brown glass vial and evaporated overnight in a nitrogen chamber at 50 °C. Subsequently, the vials were stored at −20 °C.

### 4.6. Separation and Analysis of Lipid Classes by High-Performance Thin-Layer Chromatography (HPTLC)

Previously extracted lipid samples were dissolved in chloroform at a concentration according to their weight and tissue type. Additionally, chromatographic standards (dissolved in chloroform) were used, assigning changes in the lipid profile to certain lipid classes. HPTLC was used to separate and visualize lipids using silica gel as stationary phase. Lipid samples and chromatographic standards were applied to a silica gel plate (Merck KGaA) using the CAMAG Automatic TLC Sampler 4 (CAMAG, Muttenz, Switzerland). Afterwards, the lipids were separated in a developing chamber using chloroform (Merck KgaA), methanol (Merck KgaA), ammonia 32% (VWR Chemicals, Radnor, PA, USA) and lipid-free water (Carl Roth GmbH) in a ratio of 161:75:5: 10 as mobile phase. To detect the internal standard TopFluor LPA, the plate was analyzed at 366 nm with the CAMAG TLC Visualizer 2 (CAMAG) and at 360 nm with the CAMAG TLC Scanner 4 (CAMAG) using a mercury lamp. Afterwards, lipids were derivatized with 10% copper-II-sulfate (Merck KgaA), 8% phosphoric acid (Carl Roth GmbH) and 5% methanol (Merck KgaA) using the CAMAG Derivatizer (CAMAG). To detect the lipid profile, the plate was heated at 120 °C for 2 h [72,73] and finally visualized at 366 nm with the CAMAG visualizer and scanned with the TLC scanner at 360 nm using the deuterium lamp. Lipid profiles were analyzed using the VisionCats program 2.4 (CAMAG) and further processed with Adobe Photoshop CC 2017 and Adobe Illustrator CC 2017 (Adobe, Inc., San José, CA, USA).

### 4.7. Mass Spectrometry

Lipids were extracted as previously described [74]. Briefly, tissues were homogenized in 1 mL H_2_O using the Stomacher microBiomaster homogenizer (Seward, Worthing, UK). After addition of the internal standard (30-pmol C17-lysophosphatidylcholine (C17-LPC), 30-pmol C15-ceramide (C15-Cer), 30-pmol C17-sphingosine (C17-Sph), 10-pmol C17-sphingosine-1-phosphate (C17-S1P), all from Avanti Polar Lipids, Alabaster, AL, USA), 200 µL 6-N HCl, 1 ml methanol and 2 mL CHCl_3_ in glass centrifuge tubes, samples were vigorously vortexed for 10 min. After centrifugation at 1900× *g*, the lower CHCl_3_ phase was collected. Extraction was repeated with an additional 2 mL CHCl_3_ and the two CHCl_3_ phases were combined and evaporated using the SpeedVac RVC 2-18 Cdplus (Christ, Osterode, Germany). Samples were resuspended in 100 µL methanol/CHCl_3_ (4:1 *v/v*) and analyzed using the Prominence high-performance liquid chromatography (HPLC) system and the LCMS-8050 mass spectrometer (Shimadzu, Duisburg, Germany). Lipids were separated on a 2.1 × 150-mm Kinetex reversed-phase C-18 column with 2.6-µm particle size (Phenomenex, Aschaffenburg, Germany) with two separate programs at 50 °C oven temperature: sphingomyelin and sphingosine-1-phosphate were separated with the following program: Start with 80% mobile phase A (0.1% formic acid in H_2_O) and 20% mobile phase B (50% acetonitrile, 50% 2-propanol), 2 min 20% B, 4 min 40% B, 4 min B curve -3, 50 min 92.5% B, 52 min 100% B, 70 min 100% B, 70.1 min 20% B, 80 min 20% B, flow rate 150 µL/min. Ceramides and sphingosine were separated with the following program: Start with 90% mobile phase A (0.1% formic acid in H_2_O) and 10% mobile phase B (acetonitrile), 10 min 25% B, 20 min 35% B, 40 min 75% B, 40.1 min 95% B, 70 min 95% B, 70.1 min 10% B, 80 min 10% B, flow rate 200 µL/min. Mass spectrometry was carried out in MRM mode under optimized conditions with the electrospray ionization (ESI) source for sphingomyelin and sphingosine-1-phosphate and the atmospheric pressure chemical ionization (APCI) source for ceramide and sphingosine. Mass transitions were as follows: C17-LPC *m/z* 510/184, C15-Cer *m/z* 524/264, C17-Sph *m/z* 286/268, C17-S1P *m/z* 366/250, Sph *m/z* 300/286 and S1P *m/z* 380/264. For the remaining analytes, the following mass fragments were quantified: ceramides, monohexosyl ceramides and lactosylceramides: *m/z* 264, dihydroceramides and monohexosyl dihydroceramides: *m/z* 266 and sphingomyelin: *m/z* 184. Data analysis was performed using LabSolutions 5.97 and LabSolutions Insight 3.1 (Shimadzu).

### 4.8. Preparation of Mouse Primary Cells

To analyze cell-specific mRNA expression levels of *S1prs* primary astrocytes, microglia, neurons and mature oligodendrocytes were prepared. For this, timed-pregnant and postnatal C57/BL6 and BALB/c mice were collected at defined times (morning of vaginal plug was considered as embryonic day 0 (E0) for preparation of primary cells). Astrocyte and microglia preparation were performed using cerebra from mice at P1-3, following the previously described protocol of Klar et al. [75]. Serum-free preparation of mouse cortical primary neurons was performed with E18 (± 0.5 days) mouse embryos as previously described by Brewer et al. [76] and Klar et al. [75]. Mature oligodendrocytes were prepared according to an adapted protocol of Chen et al. [77] and Suckau et al. [78] using cerebra from pups at P5. Purity of primary cell preparation was proven with cell-type-specific markers using qRT-PCR (Appendix A).

### 4.9. Preparation and Cultivation of NPC1-Deficient Skin Fibroblasts Derived from Patients

Written informed consent was obtained from each patient after counseling and explanation of possible consequences of the study. The study adhered to the tenets of the Declaration of Helsinki and was approved by the local medical ethics committee (Hannover Medical School, Germany (2576-2015) and Faculty of Medicine and Health Sciences at the Carl-von-Ossietzky University Oldenburg, Germany (2018-097)). Patients are siblings and both diagnosed by Morbus Niemann–Pick type C1 (NPC, OMIM257220). Molecular genetic testing of all 25 exons of the *Npc1* gene (NM_000271.5) detected the homozygous sequence variant c.3182T>C (p.I1061 T), known to cause NPC1 in humans [21,79]. The siblings (sibling 1:26 years, sibling 2:24 years) were diagnosed at the age of twelve, showing to date symptoms like vertical gaze palsy, splenomegaly and progressive ataxia with pyramidal and extrapyramidal signs (Figure 6). Currently, sibling 2 is less affected than sibling 1. Treatment of both patients with 3 × 200 mg miglustat started immediately after diagnosis. For further experiments, fibroblasts from both patients were prepared with the following procedure: After local anesthesia, a 2–3 mm skin biopsy sample from the inner arm area of each patient and control was obtained and cultured as previously described by Villegas et al. [80]. In brief, skin biopsies were transferred to minimum essential medium (MEM) with 20% fetal bovine serum (FBS), 1.3% l-glutamine and 0.8% antibiotic–antimycotic (all from BioWest, Nuaillé, France). Biopsies were cut into pieces of approximately 1 to 2 mm^2^ and transferred to a sterile 75-cm^2^ culture flask. The skin explants were briefly left to air dry and to attach to the bottom of the culture flasks. Then culture medium was added and the skin explants cultured for 7 to 14 days at 37 °C and 5% CO_2_. After primary fibroblasts reached confluency of 50% to 75%, cells were trypsinized and transferred to a new 75 cm^2^ flask for maintenance and experiments [80,81]. After the splitting process, 1, 5, 10 and 15 (p1, p5, p10, p15) RNA was extracted from about 500,000 cells for further qRT-PCR analyses. To verify the cholesterol accumulation, filipin staining was performed (Appendix A). Due to the lack of fibroblasts from healthy family members, fibroblasts from an apparently healthy individual (GM08398) obtained from Coriell Institute for Medical Research (Camden, NJ, USA) were used as a control for filipin staining. Fibroblasts were cultured in high glucose Dulbecco’s modified Eagle’s medium (DMEM) + GlutaMAX^TM^-I (Life technologies/Gibco, Carlsbad, CA, USA), supplemented with 10% FBS (Life technologies/Gibco) and 1% penicillin/streptomycin (Thermo Fisher Scientific, Schwerte, Germany) at 37 °C and 5% CO_2_.

### 4.10. Derivation of Neuronal Cells from Human Induced Pluripotent Stem Cells (iPSCs)

Human iPSCs were differentiated into a mixed culture of neurons and glia cells. Reprogramming of NPC1 patient-specific fibroblasts, carrying the *Npc1* homozygous mutation c.3182 T>C, [p.I1061T] (GM18453, Coriell Institute for Medical Research) and fibroblasts of an apparently healthy control individual (GM05659, Coriell Institute for Medical Research), is described in detail in Peter et al. [82] and Trilck et al. [83], respectively. Neural differentiation of iPSCs was started by induction of the formation of neural rosettes. Once neural rosettes had spontaneously formed, neural progenitor cells were isolated using magnetic beads coated with antibodies against the marker of the neural lineage PSA-NCAM (Miltenyi Biotec, Bergisch Gladbach, Germany). The neural progenitor cells obtained were expanded and maintained in proliferation medium containing DMEM (Thermo Fisher Scientific), 40% DMEM/F-12 (Thermo Fisher Scientific), 1X B27 (Thermo Fisher Scientific), 0.5% penicillin/streptomycin (Thermo Fisher Scientific), 20-ng/mL basic fibroblast growth factor (FGF2, Amsbio, Abingdon, United Kingdom), 20-ng/mL epidermal growth factor (EGF, PeproTech, Hamburg, Germany). Terminal differentiation of neural progenitor cells was initiated by using differentiation medium containing DMEM, 40% DMEM/F-12, 1X B27, 0.5% penicillin/streptomycin, which was changed every 4 days over a period of 40 days. For detailed protocols see Peter et al. [82] and Trilck et al. [83].

### 4.11. RNA Extraction and cDNA Synthesis

RNA was extracted from the following samples: brain tissue of sham-treated and treated *Npc1^+/+^* and *Npc1^-/-^* mice, mouse primary cells, NPC1 patient-specific fibroblasts (sibling 1, sibling 2, GM18453) and fibroblasts of an apparently healthy individual (GM05659). Samples were homogenized with TRIzol reagent in accordance with the manufacturer’s recommendation (Thermo Fisher Scientific, Waltham, MA, USA). Brain tissue was further homogenized mechanically using a syringe and needle. The further extraction process was performed according to the manufacturer’s protocol. Isolated RNA was dissolved in an appropriate amount of ultrapure water. The RNA concentration was determined by measuring the absorbance at 260 nm with the BioSpectrometer basic (Eppendorf). RNA samples were stored at −80 °C until further use. cDNA synthesis was performed using the high-capacity cDNA reverse transcription kit (Thermo Fisher Scientific). The 2–5 µg RNA was transcribed to cDNA at a final concentration of 0.1 µg/µL. Control reactions were performed without MultiScribe reverse transcriptase (Thermo Fisher Scientific). cDNA was stored at −20 °C. The quality of the cDNA was proven via PCR analysis with *Actb* primers: ms *Actb* 5’- cac agc tga gag gga aat cgt gcg tga -3’ and 5’- tgc ggt gca cga tgg agg ggc cgg act -3’, hm *Actb*: 5’- ccg agc ggg aaa tcg tgc gtg a -3’ and 5’- ggg ccg gac tcg tca tac tcc t -3’. The following thermal cycling parameters were used: 1 cycle with 95 °C for 2 min; 30 cycles with 95 °C for 30 s, 68 °C (hm *Actb primers*)/70 °C (ms *Actb primers*) for 30 s and 72 °C for 1 min; 1 cycle with 72 °C for 5 min. cDNA was subsequently used for qRT-PCR analyses.

### 4.12. Quantitative Real-Time PCR (qRT-PCR)

qRT-PCR analysis was performed to detect *S1pr1-5* mRNA expression in different brain regions of sham-treated and treated *Npc1^+/+^* and *Npc1^-/-^* mice (Figure 3), in mouse primary cells (Appendix A), NPC1 patient-specific fibroblasts (sibling 1 and sibling 2 shown in Figure 6, GM18453 shown in Appendix A,) and fibroblasts of an apparently healthy individual (GM05659 shown in Appendix A). Furthermore, qRT-PCR analysis was also used to check the purity of the prepared primary cells using specific cell markers shown in Appendix A (*Gfap* for astrocytes, *Iba1* for Microglia, *Tuj1* for neurons, *Ng2* for immature oligodendrocytes and *Mbp* for mature oligodendrocytes). The TaqMan^®^ Fast Universal PCR Master Mix (2X), No AmpErase^®^ UNG (Thermo Fisher Scientific) was used for qRT-PCR. According to the manufacturer’s protocol, each reaction contained 100 ng cDNA, 1 µL of appropriate TaqMan^®^ expression assay (listed in Appendix A, Thermo Fisher Scientific), 9 µL ultrapure water and 10 µL TaqMan^®^ Fast Universal PCR Master Mix (2X), No AmpErase^®^ UNG. All TaqMan^®^ expression assays consists of a TaqMan™ Probe (5 µM), a forward primer (18 µM) and a reverse primer (18 µM). The TaqMan™ Probe is labeled with the reporter dye FAM (6-carboxyfluorescein) at the 5’- end and with MGB (minor groove binder) attached to a non-fluorescent quencher (NFQ) at the 3‘- end. The following thermal cycling parameters were used: 95 °C for 20 s, 95 °C for 1 s and 60 °C for 20 s for 45 cycles. qRT-PCR was carried out at the 7900 HT Fast real-time PCR System using SDS 2.3 software (Applied Biosystems, Foster City, CA, USA), the ViiA 7 real-time PCR System using the QuantStudio real-time PCR software 1.2. (Thermo Fisher Scientific) or on the CFX96 Touch™ real-time PCR detection system using CFX Manager Software 3.1 (Bio-Rad Laboratories, Hercules, CA, USA). Expression data were normalized to *Ppia*, *β-Actin* and/or *Gapdh*, that were proven as useful reference genes for qRT-PCR [84,85,86]. To analyze the relative change in mRNA expression, the 2^-ΔCt^ method was used [87]. For each reaction, two replicates were performed. All data show the mean ± SEM of at least three independent experiments.

### 4.13. Cloning of EGFP-Coupled mS1pr1-5 Constructs

For HEK293H cell transfection and subsequent western blot analyses, the following expression plasmids were generated: pEGFP-N1-*mS1pr1*, pEGFP-N1-*mS1pr2*, pEGFP-N1-*mS1pr3*, pEGFP-N1-*mS1pr4* and pEGFP-N1-*mS1pr5*. *mS1pr1* cDNA (GenBank accession no: NM_007901) was subcloned from DDK-Myc-tagged pCMV6-entry (MR205968, OriGene, Rockville, MD, USA) into pCR™2.1-TOPO™ Vector (Thermo Fisher Scientific, Waltham, MA, USA) by PCR amplification using oligonucleotides (synthesized from Eurofins genomics, Ebersberg, Germany) to insert restriction sites instead of the stop codon (listed in Appendix A). The PCR product was purified with the QIAEX gel extraction kit (Qiagen, Hilden, Germany). *mS1pr2* ORF, *mS1pr3* ORF, *mS1pr4* ORF and *mS1pr5* ORF (GenBank accession no: *mS1pr2*, NM_010333; *mS1pr3*, NM_010101; *mS1pr4*, NM_010102; *mS1pr5*, NM_053190) were subcloned from pFLAG-CMV™-1 expression vector (Sigma-Aldrich) into pCR™2.1-TOPO™ Vector in the same way. Afterwards, all *S1pr* fragments were cloned into the pGFP-N1 expression vector (Clontech Laboratories, Inc, Mountain View, CA, USA) using the following restriction enzymes (Appendix A): *mS1pr1*/*mS1pr3*/*mS1pr5*: *Bam*HI, *Hind*III; *mS1pr2*: *Eco*RI, *Bam*HI; *mS1pr4*: *Hind*III, *Pst*I. Inserted *S1pr1-5* cDNA sequences were analyzed by digestion with the respective restriction enzymes (Appendix A) and sequencing, conducted at Eurofins genomics.

### 4.14. Cell Culture and Transfection of HEK293H Cells

HEK293H cells were obtained from Thermo Fisher Scientific and routinely cultivated in DMEM (PAN-Biotech, Aidenbach, Germany) supplemented with 10% FBS (PAN-Biotech), 2-mM glutamine (Merck KgaA) and 1-U/mL penicillin/streptomycin (PAN-Biotech) at 37 °C and 5% CO_2_. The following plasmids were used for transfection: pGFP-N1-*mS1pr1*, pGFP-N1-*mS1pr2,* pGFP-N1-*mS1pr3,* pGFP-N1-*mS1pr4,* pGFP-N1-*mS1pr5.* For immunocytochemistry, 1 × 10^5^ HEK293H cells were seeded on 0.1-mg/mL poly l-lysine (PLL) coated coverslips in 12 well plates and cultivated 24 h. Cells were transfected using DNA-calcium phosphate co-precipitation; 1 µg plasmid, diluted in 25 µL sterile water, 2.5 µL CaCl_2_ and 50 µL HEPES buffer (pH 7.05) were mixed, added to the wells and incubated for about 20 h. For protein analysis, 8 × 10^6^ cells were seeded at a density of 3.5 × 10^4^ cells/cm^2^ in 10 cm petri dishes and cultivated 24 h. Adherent cells were transfected with 80 µg of respective plasmid DNA diluted in 1350 µL water (cell culture grade), 150 µL CaCl_2_ (2.5 M) and 1348 µL HEPES buffer (pH 7.05) and incubated for about 20 h.

### 4.15. Immunocytochemistry

Transfected HEK293H cells were fixed with ice-cold 4% paraformaldehyde (PFA, Merck KgaA) in 1x phosphate buffer saline (PBS) containing 15% sucrose for 20 min at RT. Afterwards, cells were permeabilized with 0.1% Triton and 0.1% sodium citrate in 1x PBS for 3 min at 4 °C and then incubated for 1 h at RT in 1x PBS with 10% fetal calf serum (FCS). Cells were then incubated overnight at 4 °C with monoclonal anti-GFP antibody (632380, Clontech Laboratories, Inc, Mountain View, CA, USA), diluted 1:2500 in 1x PBS with 5% FCS. Then, goat anti mouse Alexa Fluor 488 conjugated secondary antibody (Invitrogen, Carlsbad, CA, USA), diluted 1:1000 in 1x PBS with 5% FCS, 1% bovine serum albumin (BSA) and 0.5-µg/mL DAPI was incubated for 90 min at RT. Finally, coverslips were mounted with Mowiol-DABCO Mounting medium (Carl Roth GmbH) and used for microscopy (Appendix A). Fluorescence images of HEK293H cells transfected with GFP-coupled *mS1pr1-5* plasmids were captured with the confocal laser scanning microscope Leica TCS SP8 (Leica, Wetzlar, Germany) using the confocal software (Leica). Images were taken using a 63× objective (oil-immersion, 1.2 NA) with the 488-nm line of an argon-ion laser. Adjustment of brightness and contrast were performed using ImageJ (NIH, Bethesda, MD, USA).

### 4.16. Lysate Preparation

For protein extraction of GFP-coupled S1PR1-5, transfected HEK293H cells were lysed in lysis buffer supplemented with 7× cOmplete protease inhibitor cocktail (Merck KgaA), 10× PhosStop (Merck KgaA) and 1-mM phenylmethylsulfonyl fluoride (PMSF, Thermo Fisher Scientific) using the µMACS GFP isolation kit (Miltenyi Biotec) according to the manufacturer’s protocol.

Protein extraction of different brain regions of sham-treated and treated *Npc1^+/+^* and *Npc1^-/-^* mice was performed adding 500 µL lysis buffer (20-mM Tris, pH 7.5, 0.25-M sucrose, 1-mM EGTA, 5-mM EDTA supplemented with 7× cOmplete protease inhibitor cocktail, 10× PhosStop and 1-mM PMSF). Tissue was homogenized mechanically with a syringe and a needle (0.60 × 60 mm/0.90 × 70 mm) and sonicated 3 × 3 sec. Afterwards, the lysates were incubated on ice for 30 min and centrifuged at 10,000× *g*, 10 min at 4 °C. The supernatant was transferred to another vial and stored at −20 °C.

### 4.17. Western Blot Analysis

Purified protein lysates of GFP-coupled S1PR1-5 were used to verify S1PR3/S1PR5 antibody specificity (Appendix A). To analyze changes of S1PR3 and S1PR5 protein expression in sham-treated and treated *Npc1^+/+^* and *Npc1^-/-^* mice protein extracts of the PC, FC, HC, CE and BS were prepared. Protein concentrations were determined using the Pierce BCA Protein Assay Kit (Thermo Fisher Scientific) according to the manufacturer’s protocol. Western blot was performed according to Velmans et al. [88], Vierk et al. [89] and Meyer et al. [35] with slight modifications. Purified protein lysates of GFP-coupled S1PR1-5 and total protein lysates (30–60 µg) of sham-treated and treated *Npc1^+/+^* and *Npc1^-/-^* mice were subjected to 10% or 12% sodium dodecyl sulfate polyacrylamide gel electrophoresis (SDS-PAGE) and subsequently transferred to a nitrocellulose membrane (Amersham Protran 0.45 NC, GE Healthcare, Boston, MA, USA). Blots were blocked for 1 h with 5% non-fat dry milk or 5% BSA diluted in Tris-buffered saline with 0.1% Tween^®^ 20 (TBST) or 5% BSA diluted in PBS with 0.1% Tween (PBST) and incubated overnight at 4 °C with the following antibodies: mouse anti-GFP (1:2500, 632380, Clontech Laboratories, Inc.), rabbit anti-S1PR3 (1:500, ab38324, Abcam, Cambridge, UK), rabbit anti-S1PR5 (1:1000; ab92994, Abcam) and rabbit anti-PPIA (1:4000; ab42408, Abcam). Secondary antibodies were sheep anti-mouse IgG (1:10,000; NA931, GE Healthcare, Boston, MA, USA) and donkey anti-rabbit IgG (1:10,000; NA934, GE Healthcare) conjugated to horseradish peroxidase in 5% non-fat dry milk or 5% BSA diluted in TBST or 5% BSA diluted in PBST. After incubation for 1 h at RT proteins were detected with Clarity™ Western ECL Substrate (Bio-Rad Laboratories) using the ChemiDoc™ imaging system (Bio-Rad Laboratories) and analyzed and quantified by using ImageLab 6.0 software (Bio-Rad Laboratories). Due to unspecific bands occurring with the S1PR3 antibody, an additional S1PR3 blocking peptide experiment was conducted (Appendix A). For this purpose, the S1PR3 antibody (1:150) was supplemented with 10x concentrated S1PR3 blocking peptide (synthesized by Thermo Fisher Scientific, dissolved in PBS) in 5% BSA diluted in TBST and incubated overnight at 4 °C with agitation. A control reaction was carried out using PBS instead of S1PR3 blocking peptide. Analysis, quantification and adjustment of brightness and contrast were performed using Image Lab software 6.0 (Bio-Rad Laboratories). Quantified western blots show the average of 4 to 5 separate experiments (*n* = 4–5), each of them was reproduced 3 times (*N* = 3).

### 4.18. Filipin Staining

NPC1 patient-specific fibroblasts (passage 5 of primary fibroblasts from sibling 1 and sibling 2) and fibroblasts of an apparently healthy individual (control: GM08398) were seeded at a density of about 25,000 cells/cm² on PLL-coated coverslips and cultivated for 24 h at 37 °C and 5% CO_2_. Afterwards, cells were fixed with 4% PFA (Merck KgaA) dissolved in 1x PBS for 15 min at RT, washed 2 × 5 min in 1x PBS and stored in 0.02% sodium azide (NaN3) at 4 °C. Cells were incubated in 0.1-mg/mL Filipin (Polysciences, Inc., Warrington, PA, USA) for 45 min, mounted with Mowiol-DABCO Mounting medium (Carl Roth GmbH) and used for microscopy (Appendix A). Differential interference contrast (DIC) images and fluorescence images were captured using the IX83 inverted imaging system with a DP80 camera (Olympus, Shinjuku, Japan). Images were taken with a 60x UplanSApo (oil-immersion, 1.35 NA) objective using the Olympus cellSens software (Olympus). Background correction and adjustment of brightness and contrast were performed using ImageJ (NIH, Bethesda, MD, USA).

### 4.19. Immunohistochemical Analysis of the Corpus Callosum Diameter

Sham-treated and treated *Npc1^+/+^* and *Npc1^-/-^* mice used to determine callosal diameter were deeply anesthetized with a mixture of 50 mg/kg body weight ketamine hydrochloride (BelaPharm GmbH & Co. KG, Vechta, Germany) and 2 mg/kg xylazine hydrochloride (Rompun^®^, Bayer HealthCare, Leverkusen, Germany). Following this, mice were perfused with 0.9% normal saline solution and 4% PFA dissolved in 0.1-M PBS. Animals were decapitated, the entire brain was dissected out and postfixed in 4% PFA for 24 h at 4 °C. Then, 30-μm-thick coronal brain sections were cut using a vibratome (VT 1000 S, Leica). Sections were mounted on Superfrost slides (R. Langenbrinck GmbH, Emmendingen, Germany) and dried overnight at 37 °C. Rehydration and antigen retrieval were achieved via microwave treatment (20 min at 800 W; Samsung Electronics GmbH, Schwalbach am Taunus, Germany). CNPase was used as a marker for myelinated fibers. For CNPase staining, sections were rinsed and incubated in a solution (3% normal goat serum + 0.1% Triton X-100 in PBS) containing the primary CNPase polyclonal antibody (1:300, Thermo Fisher Scientific) at 4 °C overnight. Rinsing the sections with PBS was followed by the application of the secondary antibody (Alexa Fluor 488-conjugated AffiniPure Goat Anti-Rabbit IgG (1:400, Jackson ImmunoResearch Laboratories, Inc., West Grove, PA, USA) in a solution containing 3% normal goat serum + 0.1% Triton X-100 in PBS for 60 minutes at RT. Thereafter, sections were rinsed with PBS, counterstained with 4′,6-diamidino-2 phenylindole dihydrochloride (DAPI, 1:10.000, Sigma-Aldrich), washed in distilled water and embedded in Mowiol (Merck KGaA). The thickness of the corpus callosum in each serial coronal section (29 sections ± 3 of each group) was measured using an Olympus BX63 microscope (Olympus) controlled by the Olympus cellSens software (Olympus). Thereafter, the mean thickness per mouse brain was calculated.

### 4.20. Morphometric Analysis of the Diameter of the Corpus Callosum

Six to nine mice of each group were deeply anesthetized with a mixture of 50 mg/kg body weight ketamine hydrochloride (BelaPharm GmbH & Co. KG, Vechta, Germany) and 2 mg/kg xylazine hydrochloride (Rompun^®^, Bayer HealthCare, Leverkusen, Germany). Following this, Bodian-fixed, paraffin-embedded brains were cut at 20 µm thickness and Nissl-stained. The thickness of the corpus callosum was measured in coronal sections at its thickest portion, at the level of the medial habenulae (in sham-treated *Npc1^+/+^* mice at Bregma -1.94 mm).

### 4.21. Statistical Analysis

Statistical evaluation of the qRT-PCR (Figure 3) and the western blot analysis (Figure 4) of different brain regions in sham-treated and treated *Npc1^+/+^* and *Npc1^-/-^* mice was carried out with a nonparametric Mann–Whitney *U-*test using SPSS statistics 22/24 (IBM, Chicago, IL, USA). Equally, this test was used for statistical analysis of patient-derived fibroblasts (Appendix A). Statistical analysis of the corpus callosum diameter was performed with GraphPad Prism 7 using one-way analysis of variance (ANOVA) followed by Bonferroni correction of multiple comparisons. All graphs and heat maps were created using GraphPad Prism 7 (GraphPad Software, San Diego, CA, USA). Data are given as mean ± SEM and considered significant if *p* ≤ 0.05 (* *p* ≤ 0.05, *** *p* ≤ 0.001).

## 5. Conclusions

Based on our molecular data, we assume that S1P metabolism plays an important role in NPC1 pathology, especially *S1pr3* and *S1pr5* seem to be involved. In contrast to the spleen, liver and the olfactory system the brain only shows weak protective treatment effects. However, side effects can be shown at the molecular level. Next to our mouse analysis, we showed that S1P metabolism is also involved in human NPC1 disease. Interestingly, expression analysis of *S1prs* in human fibroblasts and human iPSC-derived NPCs and NDCs showed the most prominent changes in *S1pr5* expression. This strengthened the assumption that *S1pr5* plays a role in the course of NPC1 disease. Nevertheless, we need more knowledge concerning differential regulation and dysregulation at the cellular and molecular level in NPC1 disease. This is crucial for a better understanding of the starting point and progress of the disease, to be able to generate more promising treatment protocols including new drugs.

## Figures and Tables

**Figure 1 ijms-21-04502-f001:**
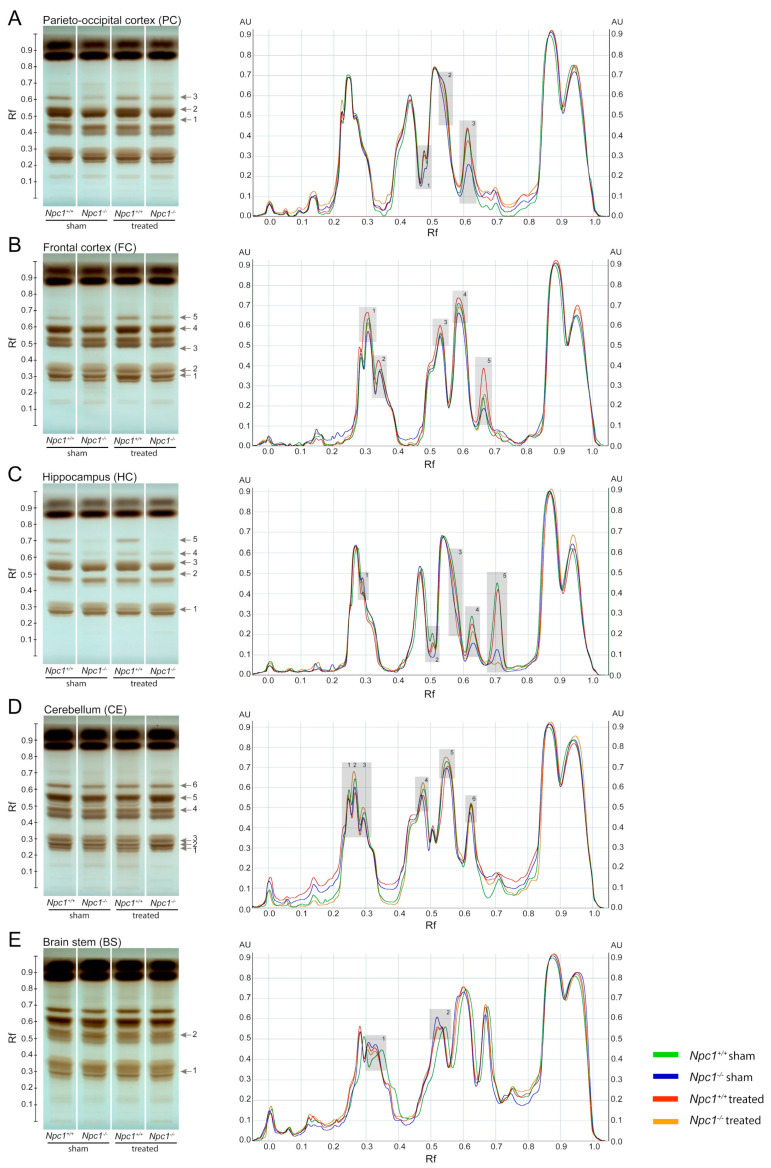
HPTLC (High-performance thin-layer chromatography) analysis from different brain tissues of sham-treated and treated *Npc1^+/+^* and *Npc1^−/−^* mice (**A**–**E**, all groups *n* = 3). HPTLC images visualized the differences of phospho- and sphingolipid amounts in the four different groups. Changed lipid classes were marked with gray arrows, corresponding to the gray boxes also shown in the digital scanning profile. Band intensities of the separated lipids were calibrated in arbitrary units (AU), representing the relative absorbance. The retention factor (Rf) represents the relative distance the substance ran compared to the distance the solvent front ran. Compared to sham-treated *Npc1^+/+^* mice, most of the lipids showed reduced band intensities in all brain regions of sham-treated *Npc1^−/−^* mice; only the brain stem showed slightly raised band intensities. The treatment of *Npc1^-/-^* mice results in most of the lipid classes to less pronounced changes of band intensities. However, some lipid classes also showed raised band intensities in treated *Npc1^+/+^* mice compared to sham-treated *Npc1^+/+^* mice. Scanning profiles were normalized by considering that the maximum on the normalization range (portion of the given track *Npc1^+/+^* sham) represents 0.9 AU. TopFluor LPA was used as loading control (see Appendix A). LPA: lysophosphatidic acid.

**Figure 2 ijms-21-04502-f002:**
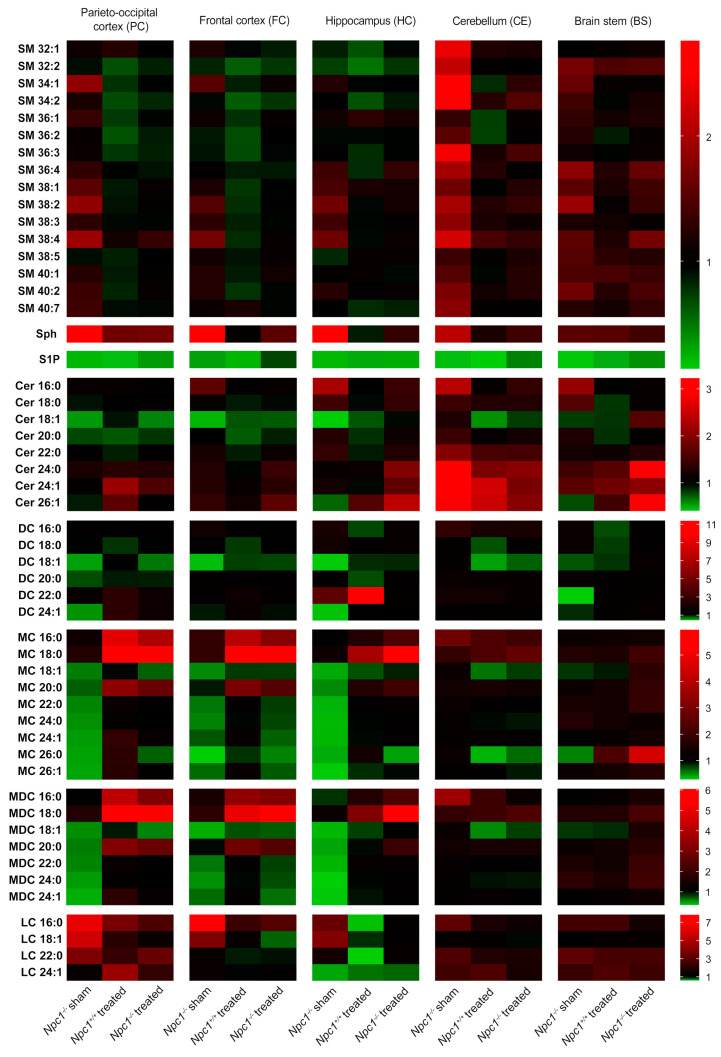
Heat map based on mass spectrometric analysis of sphingolipids from different brain regions of sham-treated and treated *Npc1^−/−^* mice and *Npc1^+/+^* mice (all groups *n* = 3). Ratio of changed intensities (arbitrary units) of the appropriate lipids are shown. Green color symbolizes lower, and red color symbolizes higher intensities compared to sham-treated *Npc1^+/+^* mice that were set to 1. Notice the lower intensity of S1P in all brain regions of sham-treated *Npc1^-/-^* mice compared to sham-treated *Npc1^+/+^* mice, whereby the treatment just marginally prevented this change. Treated *Npc1^+/+^* mice also showed reduced intensities of S1P compared to the sham-treated *Npc1^+/+^* mice. In the PC, FC and HC, some lipids (MC, MDC, DC, S1P) tended to reduced intensities in sham-treated *Npc1^−/−^* mice compared to sham-treated *Npc1^+/+^* mice. The CE showed different changes in some lipid classes, such as increased intensities in SM and Cer. The treatment prevented some changes, depending on the region and the lipid class (e.g., S1P and Cer in CE–treatment effective; S1P in PC, HC and BS–treatment not effective). All data represent the mean of *n* = 3 animals for all groups. Data for SM were normalized to C17-LPC. Sph was normalized to C17-Sph and S1P was normalized to C17-S1P. Data of all ceramides were normalized to C15-Cer. LPC: lysophosphatidylcholine, SM: sphingomyelin, Sph: sphingosine, S1P: sphingosine-1-phosphate, Cer: ceramide, DC: dihydroceramides, MC: monohexosyl ceramides, MDC: monohexosyl dihydroceramides, LC: lactosylceramides.

**Figure 3 ijms-21-04502-f003:**
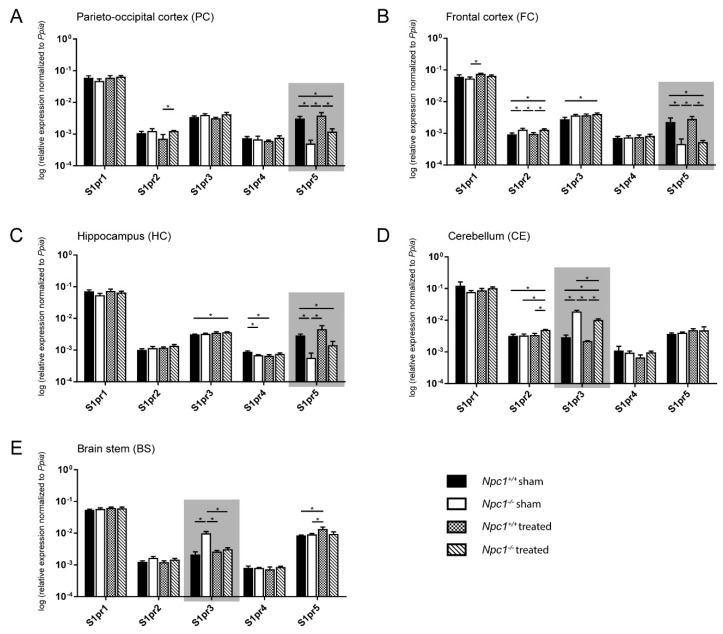
Quantitative RT-PCR analysis of sham-treated and treated *Npc1^+/+^* and *Npc1^-/-^* mice revealed a distinct change in mRNA expression of *S1pr3* and *S1pr5* in different brain regions (marked by gray boxes). In the PC (**A**), the FC (**B**) and the HC (**C**), *S1pr5* mRNA was significantly decreased in sham-treated *Npc1^-/-^* mice, whereas *S1pr3* mRNA was significantly increased in the CE (**D**) and the BS (**E**). Only in the BS, the changes of *S1pr3* expression are prevented by treatment. All other brain regions with changed mRNA levels of *S1pr3* and *S1pr5* only showed tendencies, which suggested fewer clear effects of the treatment on the transcript level. Other *S1prs,* however, showed significant changes in different brain regions (e.g., *S1pr2* in FC and CE). Data are normalized to *Ppia* and are represented as mean ± SEM with n = 3 in all groups; *p* ≤ 0.05 was considered to be significant (**p* ≤ 0.05). Statistical analysis was performed using the two-tailed nonparametric Mann–Whitney *U* test. Raw data and *p*-values are listed in Appendix A.

**Figure 4 ijms-21-04502-f004:**
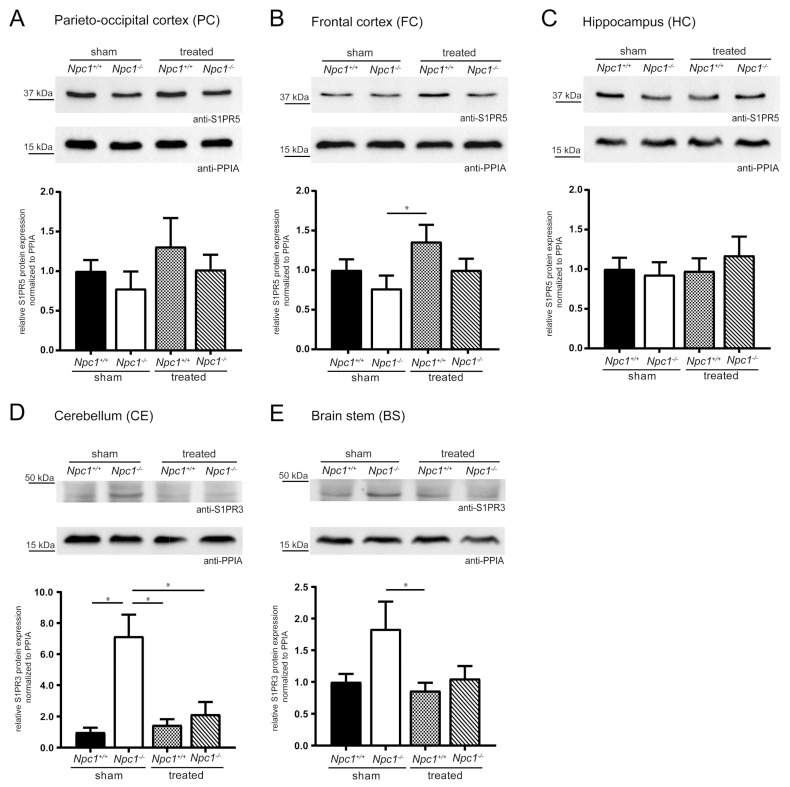
Semiquantitative western blot analyses of various brain regions from sham-treated and treated *Npc1^+/+^* and *Npc1^-/-^* mice to detect S1PR3 and S1PR5. Membrane fractions of total protein lysates were probed with anti-S1PR3 (42 kDa) or anti-S1PR5 (42 kDa) antibody. S1PR5 protein expression was examined in PC (**A**), FC (**B**) and HC (**C**). PC and FC tended to reduced S1PR5 protein expression in sham-treated *Npc1^-/-^* mice, compared to sham-treated *Npc1^+/+^* mice. This change tended to be prevented in treated *Npc1^-/-^* mice. However, treated *Npc1^+/+^* mice showed a tendency of an increase compared to sham-treated *Npc1^+/+^* mice. In contrast, in the HC, weak changes of sham-treated *Npc1^-/-^* mice could be observed. Nevertheless, treated *Npc1^-/-^* mice showed increased S1PR5 protein expression compared to the respective sham-treated mice. The S1PR3 protein expression was examined in the CE (**D**) and BS (**E**). The CE showed a significant increase in sham-treated *Npc1^-/-^* mice, compared to sham-treated *Npc1^+/+^* mice, which was prevented by the treatment, showing a significant reduction in treated *Npc1^-/-^* mice compared to sham-treated *Npc1^-/-^* mice. The BS showed the same changes as the CE. However, there was a small tendency of increased S1PR3 expression in sham-treated *Npc1^-/-^* mice compared to sham-treated *Npc1^+/+^* mice, that tended to be prevented by treatment, showing a decrease of treated *Npc1^-/-^* mice compared to sham-treated *Npc1^-/-^* mice. Protein levels were quantified and normalized according to the levels of PPIA (18 kDa). Experiments were repeated with *n* = 4 to 5 biologic controls and *N* = 3 technical controls. Data represent the ratio of the mean ± SEM (sham-treated *Npc1^+/+^* set to 1). *p* ≤ 0.05 was considered to be significant (**p* ≤ 0.05). Statistical analysis was performed using the two-tailed nonparametric Mann–Whitney *U* test.

**Figure 5 ijms-21-04502-f005:**
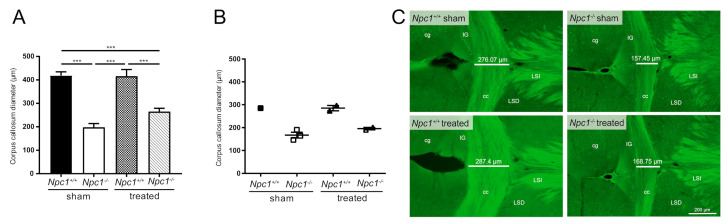
Morphometric analysis of the corpus callosum diameter from Nissl-stained brain sections of sham-treated and treated *Npc1^-/-^* and *Npc1^+/+^* mice (**A**). Sham-treated *Npc1^-/-^* mice showed significantly reduced corpus callosum diameter compared to the respective *Npc1^+/+^* mice. Interestingly, the treatment of *Npc1^-/-^* mice tended to show a slight protection effect on the thickness of the corpus callosum, that is, however, not significant. Statistical analysis was performed using one-way analysis of variance (ANOVA) followed by Bonferroni correction of multiple comparisons (*Npc1^+/+^* sham: *n* = 10; *Npc1^-/-^* sham, *n* = 9; *Npc1^+/+^* treated: *n* = 6; *Npc1^-/-^* treated: *n* = 6). Data are given as the mean ± SEM and considered significant if *p* ≤ 0.001 (****p* ≤ 0.001). Immunohistochemical analysis of the corpus callosum diameter (**B**–**C**) were performed to investigate the extent to which the reduction of the corpus callosum diameter was correlated with impaired myelination of nerve fibers (see Bräuer et al.) [34] and thus oligodendrocytes. CNPase, a marker for oligodendrocytes and myelinated fibers, were used for immunostaining. Interestingly, the measurement the corpus callosum diameter (**B**) sowed, in accordance with the appropriate immunohistochemical images (**C**) the same tendencies as the previously described morphometric analysis of the corpus callosum. The corpus callosum diameter was measured on each serial section per mouse brain (29 sections ± 3 of each group). Data are given as mean ± SEM. No statistical test was performed due to the small number of mice involved (*Npc1^+/+^* sham: *n* = 1; *Npc1^-/-^* sham: *n* = 3; *Npc1^+/+^* treated: *n* = 2; *Npc1^-/-^* treated: *n* = 2). Scale bar in the right lower picture, in the right lower corner: 200 µm. cg: cingulate cortex; cc: corpus callosum; IG: indusium griseum; LSD: dorsal lateral septum; LSI: intermediate lateral septum.

**Figure 6 ijms-21-04502-f006:**
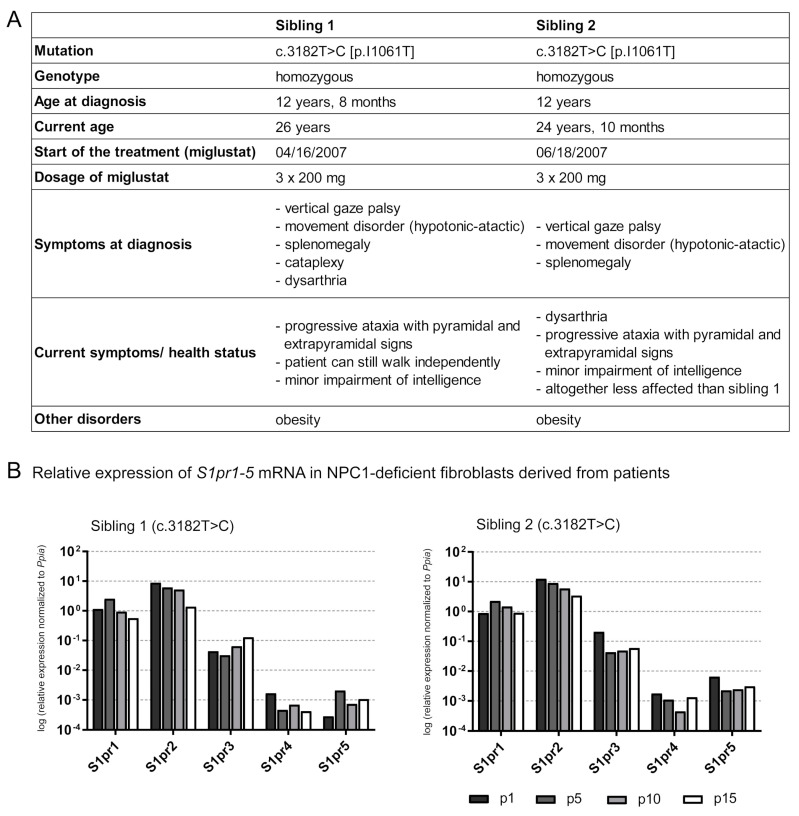
Characterization of the NPC1 disease of two siblings carrying the homozygous mutation c.3182 T>C (**A**). Both patients show neurological manifestations, such as ataxia, as well as visceral symptoms (splenomegaly). Fibroblasts of both siblings were used for further experiments: qRT-PCR was performed (**B**), suggesting that multiple washing steps could change the miglustat concentration in fibroblasts. *S1pr1-5* mRNA expression in NPC1-deficient fibroblasts were analyzed over 15 splitting procedures (p). Comparing the expression at p15 to that at p1, sibling 1 tended to reduce *S1pr1*, *S1pr2* and *S1pr4* mRNA and increased *S1pr3* and *S1pr5* mRNA level. Sibling 2 reveals similar changes in *S1pr1* and *S1pr2* expression. Comparing the expression of p15 to p1, however, sibling 2 also showed decreased expression of *S1pr3*, *S1pr4* and *S1pr5*. Interestingly, *S1pr5* expression level was clearly decreased in sibling 1, the symptomatically more affected patient, compared to sibling 2. *N* = 1 technical control, normalized to *Ppia*. p1, p5, p10, p15 = passaging number of fibroblasts.

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
