# Peer review of "Identification of Brain-Specific Treatment Effects in NPC1 Disease by Focusing on Cellular and Molecular Changes of Sphingosine-1-Phosphate Metabolism"

_ijms, 2020, doi:10.3390/ijms21124502_

Round 1

Reviewer 1 Report

The combined therapy of HPßCD, miglustat and allogrenanolone has been tested in the NPC1-/- mice (a well-characterized model for Niemann Pick type C disease) by different groups (Davidson et al., 2009, PloS One 4: e6951; Hovakimyan et al., 2013, Neuroscience, 252:201-211) which demonstrated its effect in extending mice lifespan and in improving their motor function but not their cognitive deficits.

In the paper from Gläser et al., the authors evaluated the effect of this beneficial combination on lipid metabolism in different brain regions of NPC1-/- mice and in human fibroblasts derived from patients. This paper can be considered the continuation of the one recently published by the same group and demonstrating that HPßCD/miglustat/allogrenanolone therapy can restore lipid homeostasis in the spleen of NPC1-/- mice and can reduce inflammation and altered immune cell numbers in this organ (Neßlauer et al., 2019, Lipids Health Dis, 18:146-164). With a similar approach, in the present paper authors analyzed lipids in parieto-occipital cortex (PC), frontal cortex (FC), hippocampus (HC), cerebellum (CE) and brain stem (BS); the results indicated that the analyzed lipid classes were differently affected by the disease and that treatment impacted on them in a very different way on the basis of the brain area and of the genotype; interestingly, for some lipids opposite effects were revealed in treated-WT compared to treated-NPC1-/- mice. In particular, sphingosine-1-phosphate (S1P) was found reduced in all the analyzed brain regions but the treatment didn’t improve its deficit in NPC1-/- mice while inducing a reduction in WT mice. The analysis of the expression of S1P receptors (S1PRs1-5) by qRT-PCR and western blotting in WT and NPC1-/- brains revealed a significantly increased level of S1PR3 in CE and BS of NPC1-/- mice that was restored to a normal expression by the treatment only in CE. When the same analysis performed in mice was extended to human cells, a different pattern of expression of S1PRs was found: S1PR5 was the most affected and it was increased in fibroblasts from patients but decreased in both iPSC-derived neural progenitor cells and neuronal differentiated cells.

COMMENTS

The present paper is interesting and well-written; it shows a huge amount of data concerning the expression pattern of lipids in brains of NPC1-/- mice and in human-derived cells and the impact of the triple combination HPßCD/miglustat/allogrenanolone on them; nevertheless, few editing corrections are necessary:

  • check the spelling of “focused” in the text: on lines 116, 181 and 241, “focussed” should be corrected in “focused”;
  • on line 201 of paragraph 2.2.3 the authors stated that “the treatment of NPC1+/+ mice showed a strong increaseof the S1P...” but the green color indicates a decreasein S1P;
  • on line 235, add the missing “e” to “Th”;
  • on line 236 of the paragraph 2.2.8 the authors stated that “…treatment of Npc1+/+ mice caused less smaller changes in the PC, FC, CE, and BS, but distinct increasesof most LC in the HC”; again, the green color indicates a decreasein LC;
  • on y axis of figure 3, add the missing “e” to “relativ”;
  • on lines 572-573, it should be specified that the observed changes in NPC protein expression and in cholesterol accumulation are due to the treatment of fibroblasts with fingolimod.

Author Response

We thank the Reviewer for the constructive Feedback. We have modified the manuscript according to the suggestions. Therefore all editing corrections are now included and marked in yellow.

Reviewer 2 Report

Based on comprehensive evidence, including supplemental data, this article analyzed the sphingosine-1-phosphate (S1P)/S1P receptor (S1PR) axis in different brain regions of Npc1-/- mice and evaluated the combined effects of miglustat, 2-hydroxypropyl-β-cyclodextrin, and allopregnanolone.  This article is well written and of interest to the readers in this field. Although not required in this article, the reviewer would like to know the effect of each of the three compounds.

Author Response

We thank the reviwer for the kind feedback. We agree that to know the effect of each of the three compounds would be very interesting. We hope to show these f results in the next mansucript.